# Spindle tubulin and MTOC asymmetries may explain meiotic drive in oocytes

Tianyu Wu[1], Simon I.R. Lane [1,2], Stephanie L. Morgan[1] & Keith T. Jones [1]

In the first meiotic division (MI) of oocytes, the cortically positioned spindle causes bivalent segregation in which only the centre-facing homologue pairs are retained. 'Selfish' chromosomes are known to exist, which bias their spindle orientation and hence retention in the egg, a process known as 'meiotic drive'. Here we report on this phenomenon in oocytes from $F_1$ hybrid mice, where parental strain differences in centromere size allows distinction of the two homologue pairs of a bivalent. Bivalents with centromere and kinetochore asymmetry show meiotic drive by rotating during prometaphase, in a process dependent on aurora kinase activity. Cortically positioned homologue pairs appear to be under greater stretch than their centre-facing partners. Additionally the cortex spindle-half contain a greater density of tubulin and microtubule organising centres. A model is presented in which meiotic drive is explained by the impact of microtubule force asymmetry on chromosomes with different sized centromeres and kinetochores.

[1] Biological Sciences, Faculty of Natural and Environmental Sciences, University of Southampton, Southampton SO17 1BJ, UK. [2] Present address: Institute for Life Sciences, University of Southampton, Southampton SO17 1BJ, UK. Correspondence and requests for materials should be addressed to S.I.R.L.(email: simon.lane@soton.ac.uk) or to K.T.J.(email: K.T.Jones@soton.ac.uk)

'Selfish' chromosomes that break Mendel's law of independent assortment, can bias their orientation on the spindle and hence retention. When this process occurs in gametes it is termed meiotic drive[1,2]. Maternal and paternal homologous chromosomes (hereafter homologue pairs) recombine during early meiosis to form a bivalent and then segregate following completion of meiosis I (MI). This reductional division can potentially generate an evolutionary tug-of-war because only one homologue pair is retained in the oocyte, while the other is lost into the polar body and subsequently degenerates. Any selective advantage that one homologue pair has for being retained over its partner will lead to a transmission ratio distortion (TRD), violating Mendel's laws of heredity that are based on random segregation. Differences in the ability of kinetochores, which are built on centromeres, to bind spindle microtubules, may help provide a molecular explanation of meiotic drive[3–5]. However, it is not understood how differential kinetochore asymmetry can actually lead to a TRD outside of special cases such as Robertsonian fusions[3], and thus there is no current mechanistic explanation of meiotic drive.

Here an $F_1$ hybrid (C57Bl/6×SJL) mouse strain is used to study meiotic drive. Two bivalents from this mouse have homologue pairs that differ significantly in their centromeric chromatin size, as measured by C-banding, as a result of parental strain differences[6]. Techniques to label these asymmetric bivalents are employed to allow their tracking in real time throughout the several hours of MI. A significant segregation bias to favour retention of the 'selfish' homologue pair is observed, which is found to be aurora kinase dependent. The meiotic spindle itself is also found to have an inherent asymmetry in its microtubule

organising centres (MTOCs) as well as tubulin density. A model is subsequently proposed in which spindle and bivalent asymmetry can be coupled, leading to meiotic drive.

## Results

**Establishment of a mouse hybrid model to study meiotic drive.** A live-cell fluorescently labelled tag of the major satellite repeat[7] was used to facilitate chromosome fate-tracking during MI in live $F_1$ hybrid (C57Bl/6×SJL) oocytes (Fig. 1a). The ratio of the major satellite repeat intensity for the two homologue pairs of each bivalent were calculated, always using the larger major satellite as the denominator. For all but two bivalents in each oocyte there existed a low level of major satellite asymmetry between the two homologue pairs of a bivalent (ratio 0.80, 95% CI: 0.74–0.86; Supplementary Fig. 1a). However, two bivalents in every oocyte displayed very noticeable major satellite asymmetry, which would be consistent with their reported C-banding. The smaller of the two bivalents had extreme asymmetry, with only a single observable major satellite repeat (Fig. 1b, c, marked '**' in b) consistent with chromosome 17, which lacks any C-banding in the parental SJL strain[6]. We confirmed that the small chromosome displaying extreme asymmetry was chromosome 17 using fluorescent in-situ hybridisation (FISH, Supplementary Fig 2). The second asymmetric bivalent (Fig. 1b, marked '*'; Supplementary Fig. 1a) was much larger, and likely to be chromosome 4, due to the reported C-band differences between parental strains. These observations were consistent in all oocytes from $F_1$ hybrids, and absent in the parental strain C57Bl/6 (Supplementary Fig. 3).

In the bivalents identified by major satellite size differences between the two homologue pairs, there were further size

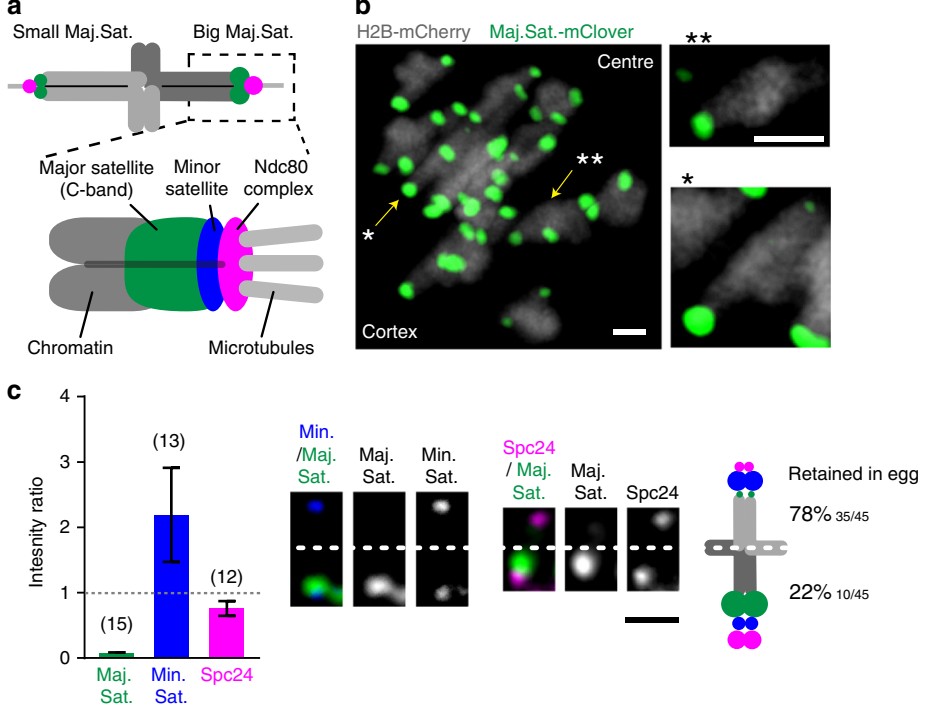

**Fig. 1** Meiotic drive in C57Bl/6×SJL $F_1$ hybrid mice. **a** Schematic showing the location of the probes used (major satellite TALE, Maj.Sat.-mClover, green; minor satellite TALE, Min.Sat.-mRuby, blue; Spc24-mCherry, magenta; chromatin, H2B-mCherry, dark grey). **b** Z-projection of all bivalents within an oocyte, showing major satellite repeats (Maj.Sat.-mClover, green) and chromatin (H2B-mCherry, grey). Insets show both driving bivalents (identified by * and **) and, readily distinguished by their unique Maj.Sat. staining. **c** Intensity ratios for major satellite (green), minor satellite (blue) and Spc24 (magenta) signals on driving bivalent with extreme major satellite asymmetry. For each ratio, the bivalent half containing the larger major satellite repeat was the denominator. Representative images of these three regions are shown, oriented with the smaller major satellite repeat above the dotted line. The schematic displays the relative sizes of the major and minor satellites and Spc24 for each half of the bivalent, and the associated percentage retention in the egg. In parenthesis, number of bivalents examined, combined from three independent experiments. Error bars are 95% confidence intervals. **b**, **c** Scale bar is 2 μm

asymmetries that could also be measured. The homologue pair of the bivalent with the larger major satellite also had consistently the smaller minor satellite and the larger kinetochore Spc24 signal (Fig. 1c; Supplementary Fig. 1b–d). Therefore, for both bivalents displaying major satellite asymmetry (* and ** in Fig. 1b) there were further asymmetries in the minor satellite and Spc24.

Using time-lapse images of oocytes at anaphase-onset with fluorescently tagged major satellite repeat we recorded the fate of these two asymmetric bivalents in each oocyte. We found both chromosomes had a significant segregation bias, such that the homologue pair with the smaller major satellite (and larger minor satellite, smaller Spc24) was retained in the egg in >75% of divisions (Fig. 1c; Supplementary Fig. 1d). Such real-time observations are consistent with an earlier report that TRD favours the chromosome with the smaller C-band[8]. From now on, because these two assymetric bivalents display the property of meiotic drive, we shall refer to them as driving bivalents.

**Driving bivalents attach with unbiased initial orientation to the meiotic spindle.** To understand how driving bivalents orientate themselves in an oocyte to favour a TRD, they were imaged from the time of nuclear envelope breakdown (NEBD) to anaphase, which is a period of several hours ($8.30 \pm 1.30$ h, mean $\pm$ std. dev., 29 oocytes). In all further studies the focus was only on the smaller sized driving bivalent (Fig. 1c). This driving bivalent alone could be reliably tracked during the several hours of prometaphase because its extreme major satellite repeat asymmetry was easy to detect in all frames.

We defined the orientation of Chromosome 17 by the direction of the large major satellite belonging to one homologue pair of the bivalent. This could be either facing the oocyte cortex (defined as 'Maj.Sat.-Cortex'), and so would be extruded into the polar body at anaphase; or as facing the oocyte centre ('Maj.Sat.-Centre'), which would result in the homologue pair being retained in the oocyte at anaphase (Fig. 2a). This orientation was defined even when the spindle was still at the centre because a time series had been captured and the future direction of travel of the bivalent with respect to the cortex known. At other times when there was no apparent tension across the bivalent, either prior to spindle formation, or during bivalent re-orientation, as happens frequently during MI[9], the orientation could not be ascertained and was recorded as 'undefined'.

Assessment of the driving bivalent when it first became biorientated showed no particular bias in its direction, with Maj.Sat.-Cortex in 41% ($n = 12/29$) of oocytes and Maj.Sat.-Centre in the remainder ('initial', Fig. 2b). However, by anaphase-onset there was a significant increase in the number of oocytes with Maj.Sat.-Cortex (76%; $n = 22/29$; $P = 0.016$, Fisher's exact test; Fig. 2b and Supplementary Movie 1). The final bivalent orientation favoured Maj.Sat.-Cortex, consistent with the earlier observation of the segregation of the larger major satellite into the polar body (Fig. 1c). This is because a cortically positioned homologue pair is destined to segregate into the polar body[10,11].

**Meiotic drive is dependent on aurora kinase activity.** Comparison of its initial and final orientation (Fig. 2b) suggest that the driving bivalent undergoes a period of rotation relative to the spindle axis during MI. Therefore, to determine if this could be directly observed, oocytes were examined between 2 and 6 h after NEBD. At this time, bivalents begin their interactions with the nascent spindle and previously have been reported to display frequent reorientations[9]. In the $F_1$ hybrid, the driving bivalent showed no propensity for rotation if the initial orientation was Maj.Sat.-Cortex. In all 12 oocytes observed with this initial state, the orientation remained fixed until the time of bivalent

separation at anaphase onset (Fig. 2c, e 'starts cortical'). However, in the 17 oocytes initially being Maj.Sat.-Centre, 59% ($n = 10/17$) were observed to rotate, assuming Maj.Sat.-Cortex by anaphase (Fig. 2d, e 'starts central'). Once the bivalent was correctly orientated to show drive no further rotations were observed ($n = 22$; Fig. 2e), suggesting that this configuration is particularly stable.

The above data suggest that the attachment of the driving bivalent to spindle microtubules is destabilised if the initial orientation does not favour the homologue pair that is the driver. Aurora kinases play an essential role in destabilising microtubule–kinetochore attachments in oocytes, either at the spindle poles (aurora kinase A)[12] or at the kinetochore (aurora kinase B/C)[13–16]. Therefore the role of this kinase family in the biased orientation of the driving bivalent was examined using a dominant negative Aurora C construct, which blocks Aurora B and C activity at the centromere[14]. Following expression of the construct, imaging of the bivalent throughout meiosis revealed that the loss of aurora kinase activity blocked re-orientation events of the asymmetric bivalent in all but one oocyte ($n = 29/30$, Fig. 2f). No change in orientation between the time of initial attachment and anaphase-onset was observed ($n = 13/30$ vs. $n = 13/30$, ns, $P = 1$, Fisher's exact test; Fig. 2b, f). A similar observation was made following addition of the pan-aurora kinase inhibitor ZM447439[17] (Supplementary Fig. 4). Collectively they show that aurora kinases, and especially aurora kinase B/C, play an essential role in meiotic drive, destabilising kinetochore–microtubule interaction to favour the biased retention of 'selfish' chromosomes.

**Driving bivalents can re-orientate before spindle migration.** For meiotic drive to occur it was hypothesised that there is likely to be both a chromosomal and a spindle component. The chromosome component, which is associated with differing centromere/kinetochore size, creates an imbalance in the composition of the bivalent. However, a spindle component is needed so that this imbalance can be transduced into a biased segregation. Since the analysis of the driving bivalent in $F_1$ mice indicated that the orientation bias was exerted between 2 and 6 h after NEBD, we looked for differences in the cortical and central halves of the spindle that existed during this time window.

Initially we wanted to examine the timings of bivalent rotation with respect to spindle migration. This is because it was hypothesised that spindle movement through the cytoplasm may generate an asymmetrical pulling force that acts on the bivalents—and such force may constitute the spindle component described above. The movement of the entire meiotic spindle to the cortex occurs during MI and can be visualised in oocytes by expressing α-tubulin-GFP and H2B-mCherry (Fig. 3a). We used this approach to examine if bivalent rotation associated with drive was occurring at the same time as spindle migration. Although previous studies have defined the timing of spindle migration, as well as details of its speed, these parameters may vary from strain to strain. As such it was important to make such measurements in the present study[18–20]. Time-lapse tracking revealed that there was little movement of the spindle in the first 6 h after NEBD (speed range: 0–3 nm min$^{-1}$, migration distance: 2–7 μm, $n = 5$), but then it moved rapidly to the cortex until MI was complete (speed range: 7–100 nm min$^{-1}$, migration distance: 10–20 μm, $n = 5$; Fig. 3b). Therefore, such timings put the events of bivalent rotation, occurring 2–6 h after NEBD (as seen in Fig. 2d, e), ahead of spindle migration to the cortex, which only becomes apparent after 6 h. Such timing do not rule out a contribution of migration to meiotic drive, as has been described very recently[21], but there must exist at least in this instance, a separate factor whose

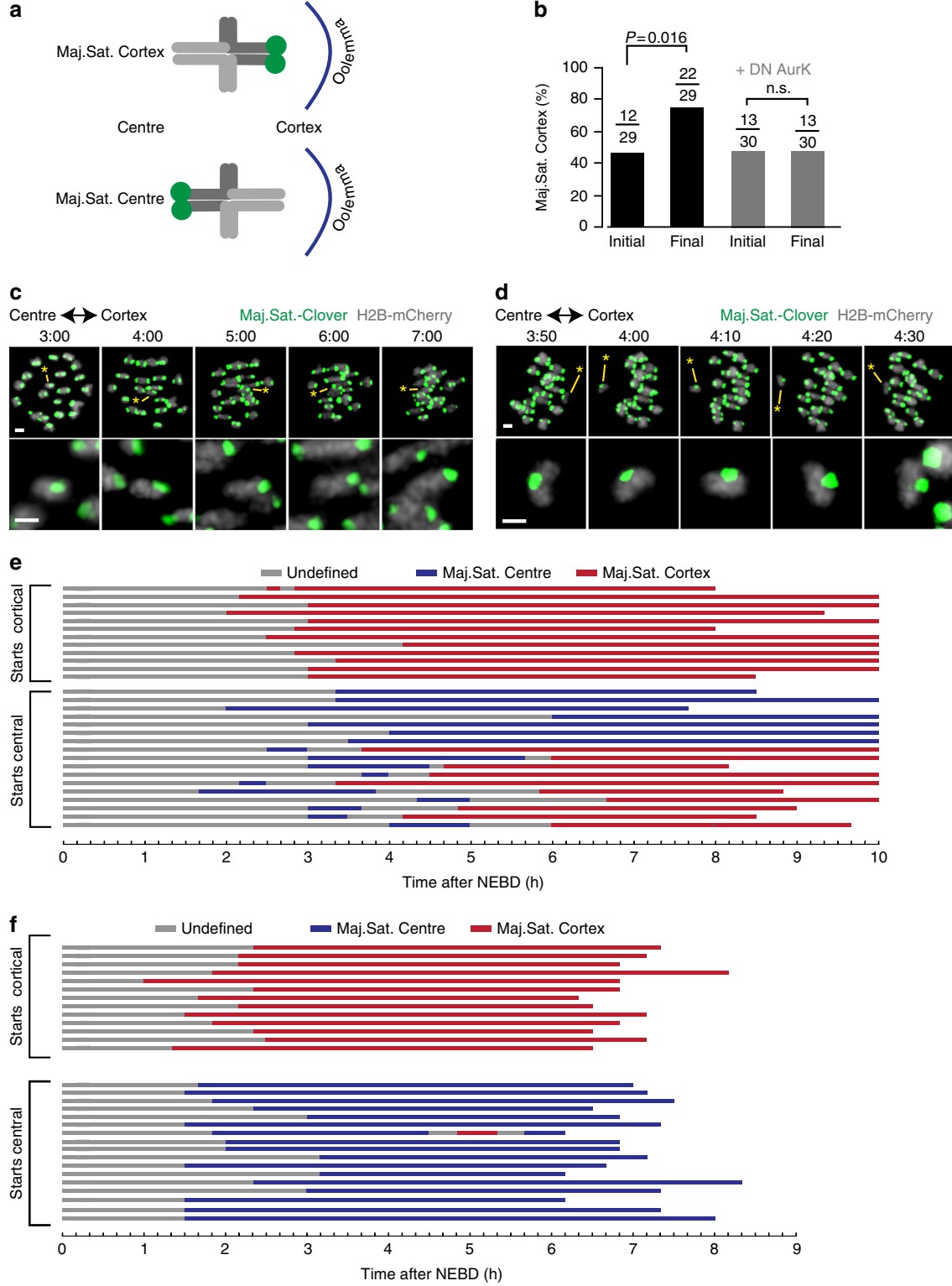

**Fig. 2** Meiotic drive requires aurora kinase activity. **a** The driving bivalent was denoted as having the large Maj.Sat. signal facing towards either the oocyte cortex (Maj.Sat. Cortex) or centre (Maj.Sat. Centre). **b** Initial and final bivalent orientation, measured at the time of first bi-orientation (initial), and the frame immediately before anaphase (final) in oocytes treated with or without the DN-aurora kinase C construct (*P* values from Fisher's exact test). Numbers of bivalents examined are given, with two independent experiments using the construct and five without. **c**, **d** Representative images of an oocyte expressing H2B-mCherry (grey) and Maj.Sat.-mClover (green), displaying the driving bivalent with persistent 'Maj.Sat. Cortex' orientation (**c**), and rotation from 'Maj.Sat. Centre' to 'Maj.Sat. Cortex' (**d**). The driving bivalent is indicated with an asterisk and enlarged in the lower image (**c**, **d**). Scale bars represent 2 μm. Times are h:mm after NEBD. **e**, **f** Time course of driving bivalent orientation (Maj.Sat. Cortex, red; Maj.Sat. Centre, blue; undefined, grey) from NEBD to anaphase or 10 h after NEBD as shown in 'c' and 'd' in the absence (**e**) or presence (**f**) of DN-aurora kinase C. Oocytes are sorted by the initial orientation of the driving bivalent (initial Maj.Sat. Cortex, top; initial Maj.Sat. Centre, bottom)

**a**

Tubulin-GFP H2B-mCherry

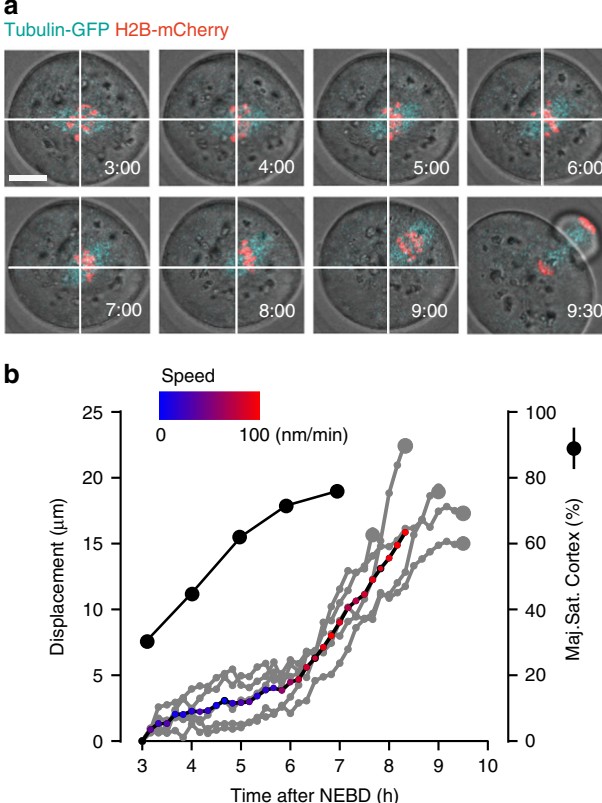

**Fig. 3** Chromosome re-orientation precedes spindle migration to cortex. **a** Bivalent migration to the oocyte cortex in a representative oocyte (H2B-mCherry, red; α-Tubulin-GFP, cyan). Times (h:mm) are relative to NEBD. White cross shows chromosome centre of mass at 3:00 h. Scale bar is 20 μm. **b** Displacement of chromosomes (grey, $n = 5$ oocytes combined from two independent experiments) between 3 and 10 h after NEBD. Displacement is relative to the starting image. The red/blue line represents the mean displacement of all bivalents, with speed colour coded as indicated (blue = 0 nm min$^{-1}$, red = 100 nm min$^{-1}$). For comparison the timing of the driving bivalent re-orientation is shown (% of bivalents Maj. Sat. cortex, $n = 29$ oocytes, black line, combined from five independent experiments). Such re-orientation of the driving bivalent preceded spindle migration. Larger grey circle shows the timing of anaphase for each oocyte

contribution can be seen while the spindle is still at the centre of the cell.

**Asymmetric bivalents stretch before spindle migration**. Despite the fact that drive can occur before spindle migration to the cortex, we still wanted to determine if there was an asymmetry in tension exerted on bivalents during MI. The desire to focus on this asymmetry was because aurora kinases destabilise microtubules at kinetochores by responding to low tension[13,22,23]. The experiments above demonstrate that if such asymmetry existed it should be evident before spindle migration.

We wished to examine the amount of tension the spindle was able to exert on bivalents, hypothesising that sister kinetochore pairs may experience different tensions depending on if they were cortical-facing or central-facing. This was a preferred model, because if such a tension asymmetry existed then this may favour a particular orientation of a bivalent with unequal sized kinetochores.

Previously we and others[13,24,25] have measured distances across the chromosome as a proxy for tension, under the assumption that the chromatin is elastic[26,27]. Here we wanted to

examine tension across each sister kinetochore pair in live oocytes, so the distance apart of the major satellite repeat and outer kinetochore was measured using Maj.Sat.-mClover and Spc24-mCherry, respectively (Fig. 4a). The peri-centromeric chromatin is also elastic and so changes in distance here reflect tension exerted at the kinetochore/centromere region[26,28]. To demonstrate that the measurement of centromere to kinetochore (C–Kt) separation can be used as a tension readout, tension was dissipated using nocodazole, which depolymerises microtubules, and monastrol, which collapses the spindle onto a single pole. For both agents, a significant reduction in C–Kt separation occurred (Fig. 4b), consistent with the measurements correlating with tension.

We next examined C–Kt separation at 4 h after NEBD, which precedes spindle migration (Fig. 3). At this time we could not define a central or cortical side of the spindle due to its very central position in the oocyte. Nonetheless, for each oocyte examined ($n = 12$) C–Kt measurements could be made for one side of the spindle versus the other, and for each bivalent the C–Kt measurement could be paired. Within individual oocytes it is apparent that across most bivalents a difference in C–Kt separation exists (Fig. 4c) and furthermore in 6 of 12 oocytes there was a significantly greater C–Kt separation in homologue pairs on one side of the spindle compared to the other (Fig. 4c, \*$P < 0.05$, paired $t$-test).

At 7 h, a time during which spindle migration occurs (Fig. 3), and when the spindle direction is apparent, the difference in C–Kt separation was still present and resembled that observed at 4 h (cortex: 0.70 μm, centre: 0.46 μm, $P < 0.0001$, ANOVA with Tukey's post-hoc test; Fig. 4d). This difference was observed for 81% of all bivalents tested (Supplementary Fig. 5). The difference in C–Kt stretch between centre and cortical facing kinetochores also persisted when actin depolymerisation was induced by cytochalasin B suggesting that actin does not contribute to C–Kt stretch (cortical: 0.68 μm, central: 0.51 μm, $P < 0.0001$, ANOVA with Tukey's post-hoc test; Fig. 4d). Such treatment did however block spindle migration because this process is actin dependent[18,29,30].

The above findings suggest that the meiotic spindle assembles at the centre of the oocyte with an inherent asymmetry, which establishes an unequal force on homologue kinetochores. During spindle migration the asymmetry is such that stretch is greater on homologue centromere–kinetochores facing the cortical pole. However, a similar level of disparity in C–Kt distance was also seen before any such migration. These measurements also suggest that the direction of spindle movement can be predicted before migration occurs, and correlates with the direction in which the homologue pairs experiencing greater C–Kt stretch are facing.

**Tubulin asymmetry across the migrating spindle**. The spindle appears to exert an asymmetrical stretch on the bivalents during MI, which is evident both when it is at the centre and also when it is migrating to the cortex. Microtubules were examined initially because they form k-fibres, which exert tension across the chromosome[31]. Therefore, α-tubulin-GFP and H2B-mCherry were expressed in GV stage oocytes (Fig. 5a), and using the H2B signal to define the centre of the spindle we measured tubulin intensity at 7 h after NEBD. Consistent with the C–Kt distance asymmetry, there was a significantly greater tubulin density on the spindle half facing the cortex of the oocyte (tubulin density ratio cortex/centre: 1.274, 95% CI: 1.217–1.332, $n = 10$ oocytes; Fig. 5b, c). Therefore, it is inferred that the greater kinetochore stretch is a result of the greater tubulin density found on the cortical pole.

The source of this tubulin asymmetry was next investigated. Meiotic spindles in oocytes have no centrioles, but instead

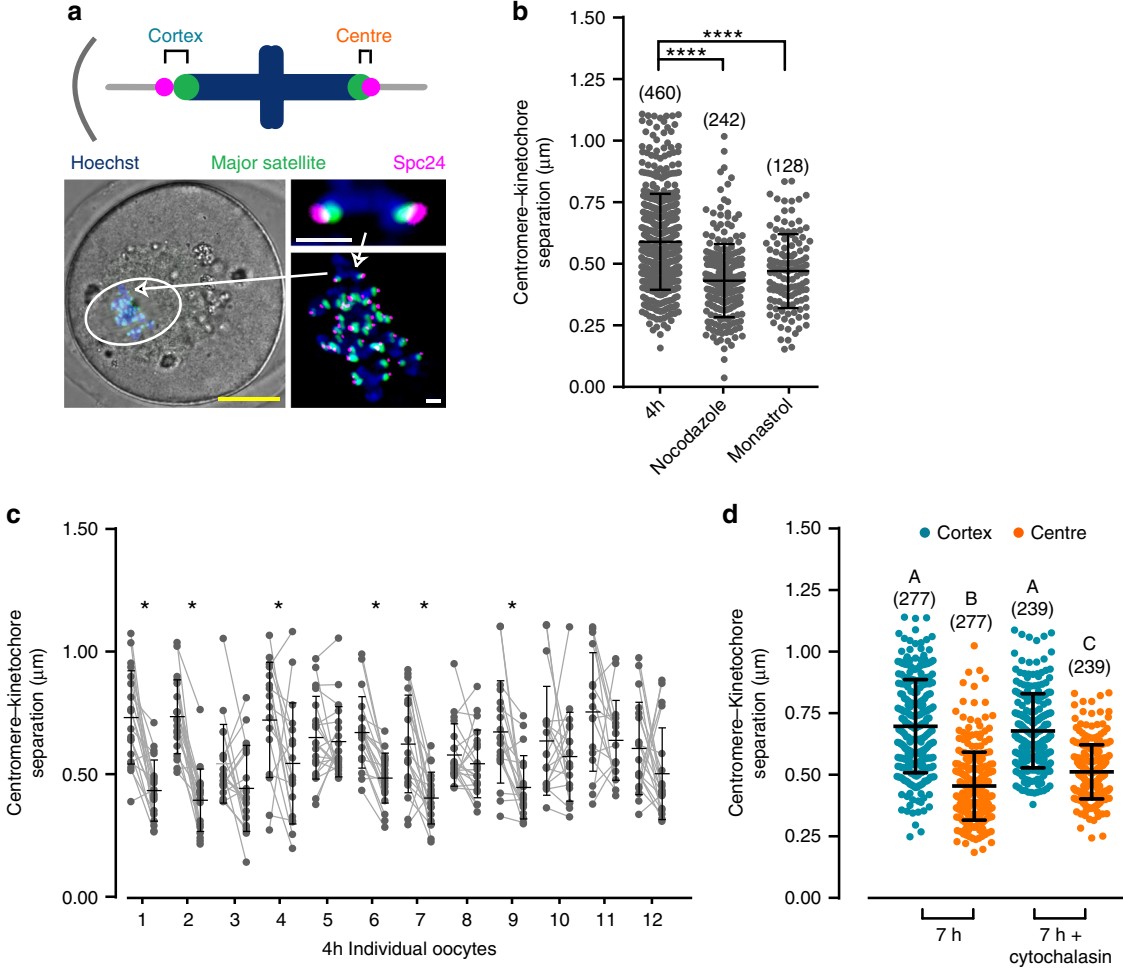

**Fig. 4** Bivalents experience greater centromere–kinetochore separation on the cortical side of the spindle. **a** Schematic to show the location of the probes used and the measurement of central and cortical centromere–kinetochore (C–KT) separation. A representative image to show centromeres (Maj.Sat.-mClover, green), kinetochores (Spc24-mCherry, magenta), and chromatin (Hoechst, blue) in oocytes. Arrow shows the same bivalent at different magnifications. Scale bars; yellow, 20 μm; white, 2 μm. **b** All C–KT separations are plotted at 4 h after NEBD following the indicated drug additions (****$P <$ 0.0001, one-way ANOVA with Tukey's post-hoc test, bivalent numbers are indicated. Measurements were made on 12, 7 and 4 oocytes with 3, 3 and 2 independent repeats, for untreated, nocodazole and monastrol groups, respectively). **c** C–KT separation distances for bivalents in oocytes 4 h after NEBD. The two centromeres of each bivalent within an oocyte are paired (connecting grey line). Bold lines indicate means and standard deviations. (*$P <$ 0.05, paired $t$-test, two-tailed). Measurements were made on 12 oocytes, combined from three independent experiments. **d** Cortical (blue) and central (orange) C–KT separations for oocytes at 7 h after NEBD, with or without prior treatment with cytochalasin B (for 1 h). Bivalent numbers are indicated, with measurements made on 12 oocytes with cytochalasin and 14 without, each combined from three independent experiments. Differences between each group were tested using one-way ANOVA with Tukey's post-hoc test; different letters denote significant difference ($P <$ 0.05). **b**, **d** Horizontal lines for each condition represent means and the error bars are s.d

microtubules are nucleated from a number of MTOCs[10,32,33]. Therefore, it seems plausible that the asymmetry in microtubule density stemmed from a similar bias in the distribution of MTOCs at the spindle poles. This idea is supported by previous observations of structural asymmetries in MTOC components in mouse oocytes during MI[34,35]. Therefore here, oocytes expressing the MTOC protein Cep192, coupled to GFP, were analysed by time-lapse microscopy during MI. When expressed in GV stage oocytes that were allowed to mature, MTOCs were first observed in the oocyte cortex, and then migrated to the nuclear envelope around the time of NEBD (Fig. 6a, b, Supplementary Movie 2). In this experiment and all others where measurable, we used GV stage oocytes that had a surrounded nucleolus (SN) configuration for chromatin (Fig. 6a) as these show the greatest potential for embryonic development, and likely represent a more mature state than the alternative non-surrounded nucleolus configuration[36].

Shortly after NEBD, MTOCs formed two major clusters that later became the spindle poles, consistent with previous reports[10,32,33]. Interestingly, as early as 1–2 h after NEBD, significant differences in MTOC density were measured at the nascent spindle poles (*$P <$ 0.05, **$P <$ 0.01, ***$P <$ 0.001, ****$P <$ 0.0001 paired $t$-test; Fig. 6c, d). These differences persisted throughout MI and could be observed both before and during spindle migration (*$P <$ 0.05, **$P <$ 0.01, paired $t$-test; Fig. 6e, f; Supplementary Movies 2, 3). From these observations it is also concluded that the direction of spindle migration was consistently toward the spindle pole containing the greater cluster of MTOCs, judged by Cep192 intensity. It is also noted that such asymmetry was not confined to MTOC component Cep192, because it could also be observed for pericentrin ($P = 0.0001$, paired $t$-test; Fig. 6g, h), which functions in MTOCs as an integral scaffold protein[37].

From the above it is clear that the asymmetry in the clustering of the MTOCs to the two spindle poles is evident at a time before

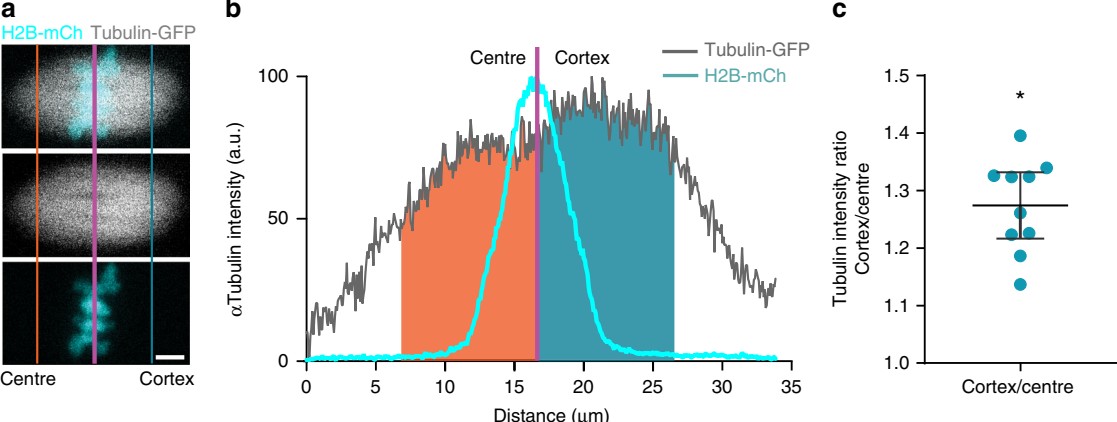

**Fig. 5** Oocyte spindle asymmetry in microtubule tubulin density. **a** Representative image of microtubules (α-tubulin-GFP, grey) and chromatin (H2B-mCherry, cyan) in an oocyte at 7 h after NEBD. Purple line indicates mean-weighted position of the bivalents; blue and orange lines indicate 10 μm distance on the cortical and central side of the bivalents, respectively. Scale bar, 5 μm. **b** α-Tubulin intensity profile, in the x-axis, of the spindle (grey) and bivalents (cyan) is shown in **a**. Total α-tubulin intensity within 10 μm of the bivalents on the central (orange) and cortical (blue) spindles halves. **c** Cortical/central ratio of α-tubulin fluorescent intensity as shown in **b** (mean 1.274). *$P < 0.05$, 95% confidence intervals shown from 1.217 to 1.332 (i.e. >1.0); $n = 10$ oocytes, combined from two independent experiments

the spindle migrates to the cortex. Finally, we co-expressed Cep192 and α-tubulin in oocytes, and labelled chromatin with Hoechst. This triple labelling, examined at 4 h after NEBD (Fig. 6i, j), allowed us to determine if tubulin asymmetry could be observed, while the spindle was at the oocyte centre and also if there was a correlation between greater Cep192 and higher tubulin density. Indeed, this was observed to be the case (Fig. 6i–k), supporting a model in which greater MTOC density nucleates a greater number of microtubules, which in turn establishes an asymmetric pull on bivalents.

## Discussion

In the hybrid mouse model studied here, we observed retention in the egg in favour of the homologue pair that contains the larger minor satellite repeat. This chromosome region is rich in CENPs[5,38], and because of the levels of these histone variants, in terms of meiotic drive, would previously have been labelled as being the 'strong centromere'[3,5,21]. The minor satellite is at the chromosomal centric constriction, and being CENP rich is the site at which the kinetochore is built[39]. Therefore the data presented here are consistent with the current idea that a 'strong centromere' favours egg retention[40].

Most of the live cell imaging analysis conducted in the present work was performed using a TALE probe directed against the major satellite, rather the minor satellite. This is because it showed a greater level of asymmetry between the two homologue pairs of the driving bivalent allowing easy identification in time-lapse imaging over the several hours of MI. The major satellite repeat is pericentric heterochromatin, rich in HP1α, which makes it DAPI dense, and suitable for C-banding. Using the TALE probe, the 'strong centromere' (more minor satellite repeat) was observed to contain a much smaller major satellite, demonstrating a reciprocity in terms of size between the two for this particular bivalent. However, being distinct but contiguous regions we are aware of no a priori size relationship that must exist, and indeed a recent study has shown size differences in the minor satellite region of chromosomes from different mice that display the same sized major satellite region[5].

The 'strong centromere' hypothesis of meiotic drive posits that a larger centromere is correlated with a larger sized overlying kinetochore. This would establish a physical asymmetry between the two homologue pairs of a bivalent in a structure, the kinetochore, having direct interaction with the spindle microtubules. Indeed such a positive size correlation between a centromere and kinetochore has previously been made with respect to the outer kinetochore component Hec1/Ndc80[3]. However, with respect to Spc24, another member of the outer kinetochore Ndc80 complex, we observe less labelling on the strong centromere. Therefore, although it remains likely that centromere size affects the overlying kinetochore structure, a simple size correlation for all 100+ kinetochore components may be too simplistic a model[39]. Indeed the relationship by which centromeric CENPs seed formation of kinetochores is currently under intense investigation[41], and involves the influence of pericentromeric heterochromatin, as well as CENP containing domains.

Additionally the minor satellite repeat is a source of non-coding RNAs that influence aurora kinase activity, which in turn is needed to destabilise erroneous microtubule attachment to kinetochores that fail to generate sufficient pulling force[42–44]. If these non-coding RNAs acted locally, in –cis, as for example Xist does on the inactive X-chromosome[45], they could then influence the rate of attachment and reattachment of microtubules to kinetochores mediated by Aurora B kinase. These other processes, in addition to kinetochore size, may have a bearing on the orientation of bivalents. Future work on the 'strong centromere' hypothesis should therefore focus on how centromere size influences recruitment of Spindle Assembly Checkpoint proteins and Chromosomal Passenger Complex components to the centromere/kinetochore, both pathways having a potential impact on the ability of the homologue pair to interact with and then destabilise spindle microtubules[46–48].

For meiotic drive to work in favour of retaining homologue pairs with a strong centromere there needs also to be meiotic spindle asymmetry. The movement of the meiotic spindle to the cortex, resulting in an unequal cell division and a polar body destined to degrade, affords the potential to generate such asymmetry. Spindle migration is associated with the building of an F-actin cortical cap, one component of which, the Rho GTPase Cdc42[18,49], was recently observed to establish tubulin asymmetry in the spindle[21]. In the present study we have measured a much earlier asymmetry, one that precedes spindle migration to the cortex and is related to the clustering of the MTOCs into two

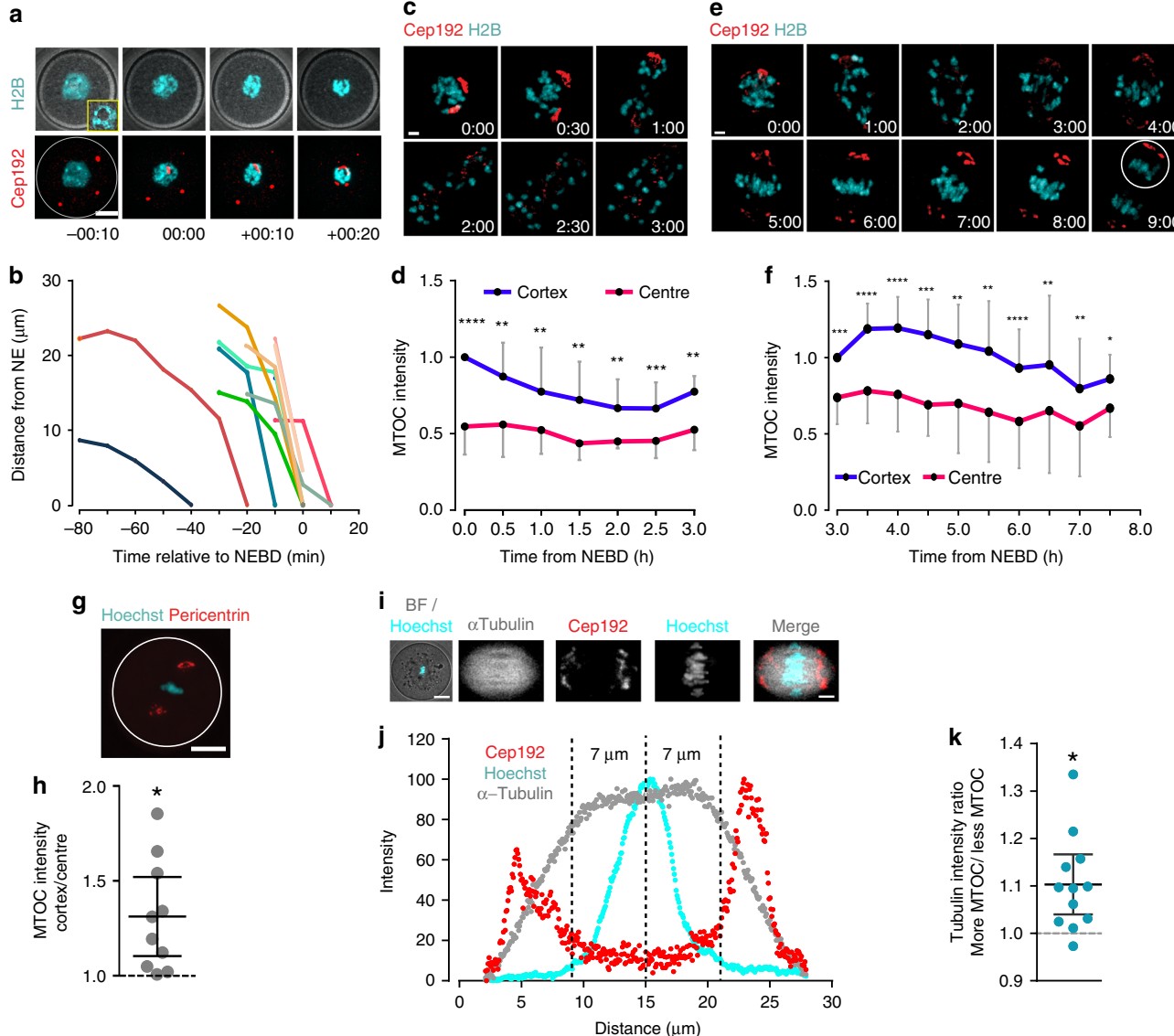

**Fig. 6** MTOC asymmetry is established at NEBD. **a, b** Representative images (**a**, Cep192-GFP, red; H2B-mCherry, cyan) and trajectory plots (**b**) of MTOCs from the oocyte cortex to centre ($n = 9$, from two independent experiments). Brightfield inset, shows single z-slice with a 'surrounded nucleolus' (SN) configuration. Time is relative to NEBD (hh:min), and distances are measured relative to the nuclear envelope. Scale bar: 20 μm. **c, d** Representative time-lapse imaging (**c**) and spindle pole intensities (**d**) of MTOCs in oocytes ($n = 9$, from three independent experiments) expressing Cep192-GFP and H2B-mCherry from 0 to 3 h after NEBD. The MTOC fluorescence intensity at both poles was calculated at 30 min intervals. Scale bar is 5 μm. **e, f** Representative time-lapse imaging (**e**) and spindle pole intensities (**f**) of MTOCs in oocytes ($n = 11$; from three independent experiments), as for (**b, c**) between 3 and 8 h after NEBD. Scale bar is 5 μm. **g** Representative z-projection of an oocyte stained for pericentrin (red) and Hoechst (cyan). White circle depicts position of plasma membrane. Scale bar is 20 μm. **h** The ratio of cortical and central MTOC intensities at 6 h after NEBD from oocytes in 'g' (oocytes from two independent experiments). **i** Representative image of a spindle (α-tubulin-GFP grey; Cep192-mCherry, red; Hoechst, cyan), at 4 h after NEBD, showing its central position within the oocyte. Scale bar: 20 and 5 μm. **j** Intensity profiles along the long spindle axis of Cep192, Hoechst and α-tubulin from the image in 'i'. Dashed lines show the weighted centre of intensity of the chromatin and 7 μm either side. **k** Ratio of the tubulin fluorescence in the regions indicated in 'j'; measurements from two independent experiments. The spindle sides are defined by having either more or less MTOC (Cep192) signal as shown in 'j'. (**d, f**) *$P < 0.05$; **$P < 0.01$; ***$P < 0.001$; ****$P < 0.0001$, paired $t$-test, two-tailed; error bars are s.d. (**h, k**) *$P < 0.05$; plots show mean line (with an $n = 10$ 'h' and 12 'k' oocytes) and error bars that are 95% confidence intervals

distinct poles after NEBD. Critically we observed bivalent rotation to favour meiotic drive at a time when the spindle had not started its journey to the cortex. While the present work does not rule out a Cdc42 contribution to meiotic drive, it shows the existence of a much earlier spindle asymmetry component, suggesting a multifactorial basis to the process. The reason for the establishment of MTOC asymmetry remains unknown, but could relate to the much earlier journey MTOCs make from the oocyte cortex to the nuclear envelope before NEBD. It may well prove that MTOCs

clustering at this time of NEBD mark an orientation for future migration of the spindle.

In summary, our data present a mechanistic basis for meiotic drive which relies on both bivalent and spindle asymmetry. In our $F_1$ hybrid model, bivalents showing meiotic drive have a smaller kinetochore, as well as differences measured in the centrosomal satellite regions. Both the size of the kinetochore and the size of the minor satellite repeat could be involved in the TRD, as kinetochores interact directly with microtubules and are

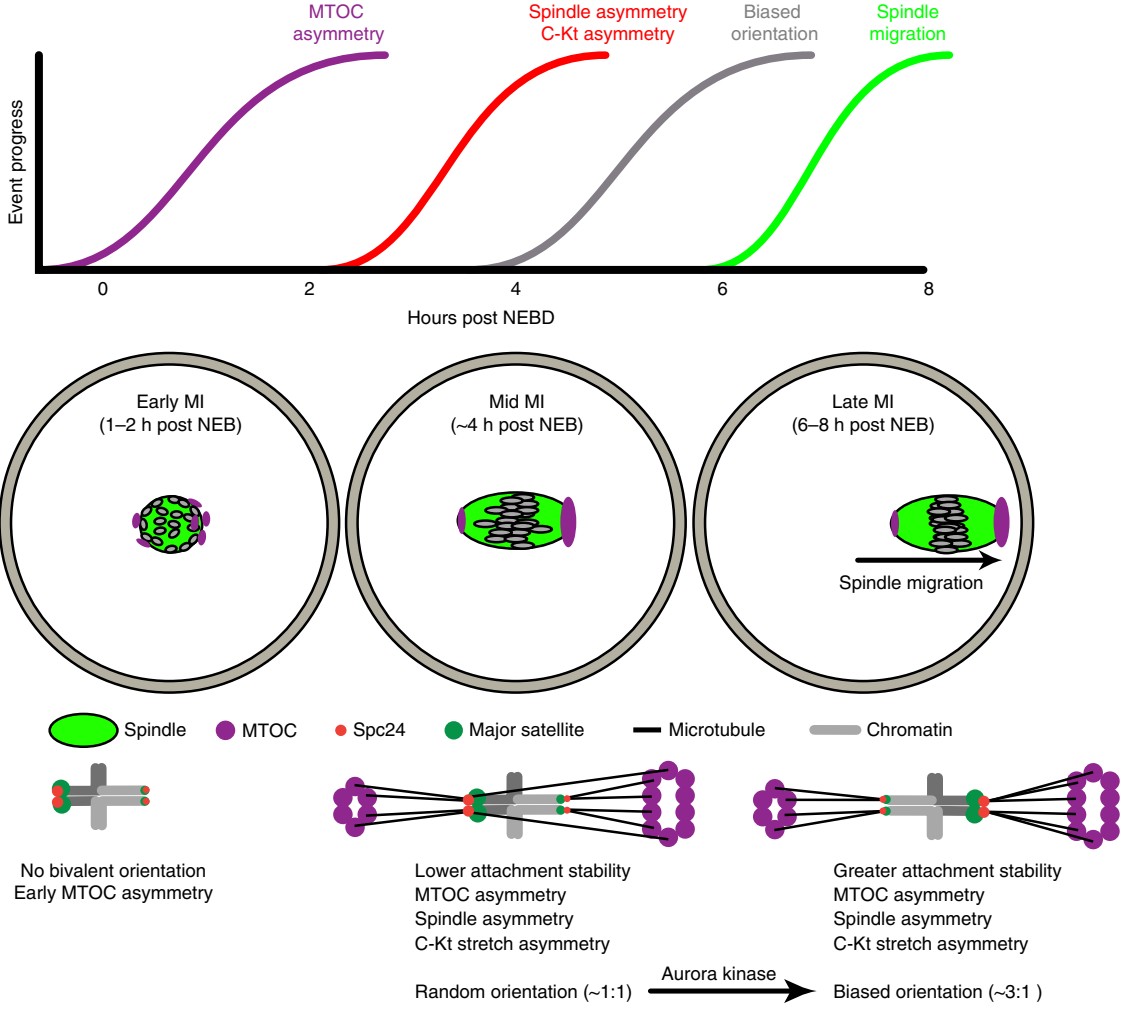

**Fig. 7** Schematic to demonstrate meiotic drive, showing the mechanism of asymmetric bivalent reorientation. The schematic shows the approximate relative timings of events relating to meiotic drive and spindle migration in times relative to NEBD (top). Cartoons graphically depict the events taking place in early, mid and late MI from left to right, respectively, in the oocyte (middle) and at the level of an individual bivalent experiencing meiotic drive (bottom)

responsible for force generation and chromosome segregation at anaphase, whilst the minor satellite repeat produces non-coding transcripts which are known to have a function in recruitment and activity of aurora kinases at the centromeres[44]. We found that biased bivalent rotation to favour meiotic drive was aurora kinase dependent, suggesting TRD uses the same mechanism as that used to destabilise erroneous microtubules attachments to kinetochores[22,46]. We suggest a model (Fig. 7) in which inherent differences in the pulling forces on kinetochores from spindle microtubules in conjunction with altered microtubule binding and destabilising abilities across asymmetrical bivalents underlie meiotic drive. The force asymmetry is likely to be derived from an asymmetry in MTOC distribution across the spindle, which is established early in MI. In such a model, larger kinetochores on the central spindle half are biased against, possibly because their interactions with spindle microtubules are never sufficiently strong to fully switch off the error correcting mechanism of aurora kinase. Indeed we have previously shown that the SAC always retains a level of partial activity during meiosis I, even during metaphase when the APC reaches its maximal activity[15].

## Methods
All reagents were from Sigma Aldrich, UK, unless otherwise stated.

**Ethics statement**. All mice were used in accordance with local and UK government regulations on the use of animals in research. This work was approved by the University of Southampton Animal Welfare and Ethical Review Board.

**Animals and oocyte culture**. Three-to-four-week female C57Bl/6 and F$_1$ (C57Bl/6xSJL) hybrid mice (Charles River, UK) were used. GV-stage oocytes were released from the ovaries 44–52 h following hormonal priming with 10 IU PMSG by intraperitoneal injection (Centaur Services, UK). Milrinone (1 μM) was added to M2 medium to maintain oocyte prophase arrest. Oocytes were stripped from the surrounding cells mechanically. For maturation, GV oocytes were washed free from milrinone and cultured in fresh M2 media.

**Microinjection**. Oocytes were microinjected in a 37 °C heated chamber (Intracel, UK) on the stage of an inverted TE300 microscope (Nikon, Japan) with micro-manipulators (Narishige, Japan)[50]. A 0.1–0.3% volume cRNA was injected using a timed pulse on a Pneumatic Picopump (World Precision Instruments, UK) using pipette RNA concentrations as follows: Cep192-GFP (600 ng μL$^{-1}$), TALE major satellite-mClover (600 ng μL$^{-1}$), Spc24-mCherry (500 ng μL$^{-1}$), H2B-mCherry (500 ng μL$^{-1}$), α-Tubulin-GFP (500 ng μL$^{-1}$). cRNAs were centrifuged for 2 min at 16,000×g before micro-injection.

**cRNA manufacture**. cRNA was transcribed in vitro from purified linear dsDNA templates. mMessage T7 or T3 RNA polymerase kits (Ambion, Life Technologies, UK) were used for the in vitro transcription reaction[24]. cRNA was suspended in nuclease-free water and the concentration of RNA products were determined by photospectroscopy. Spc24-mCherry was made by PCR from testis cDNA and restriction enzyme cloning into pRN3 derivative plasmid with C-terminal mCherry. Maj.Sat.-mClover and Min.Sat.-mRuby were gifts from Maria-Elena Torres-Padilla (Addgene plasmid #47878 and #47880, respectively)[7,51], which bind

to the major and minor satellite DNA repeats directly. Cep192-GFP was a gift from Melina Schuh[32]. α-Tubulin-GFP was a gift from Marie-Helene Verlhac[52]. Dominant negative Aurora Kinase C was a gift from Karen Schindler[14].

**Time-lapse imaging**. Timepoints were acquired at 10 min intervals using a Leica SP8 fitted with hybrid detectors, an environmental chamber set to 37 °C, and either a ×40/1.3NA or 63×/1.4NA plan apochromat oil immersion lens. In-lab software written in Python language was used to image multiple stage regions and to track chromosomes in up to 30 oocytes per experiment, using H2B-mCherry signal to ensure bivalents remained in the centre of a ~32 × 32 × 32 μm imaging volume[53].

**Inhibitor treatment**. Cytochalasin B (1 μM) was added to oocytes 6 h after NEBD. Monastrol (1 μM) and nocodazole (400 nM) were added to oocytes at 4 h after NEBD, 30 min before imaging. ZM447439 (10 μM; Tocris, USA) was added at 2 h after NEBD. All drugs were dissolved in dimethylsulphoxide and used at dilutions of 0.1% or below.

**Centromere–kinetochore separation**. Germinal vesicle stage oocytes were microinjected with mRNA encoding Spc24-mCherry and Maj.Sat.-mClover. Oocytes were matured in M2 media to 4 or 7 h after NEBD, and then counterstained with Hoechst (20 μg mL$^{-1}$). Confocal image stacks were taken with a z-separation of 300 nm and X and Y pixel size of 36 nm. Only oocytes where the long axis of the spindle was in the x–y plane were included such that centromere–kinetochore separation could be measured accurately. Core position of major satellite repeats and Spc24 were determined following Gaussian blur application ($\Sigma = 2$) using ImageJ software (NIH, USA). The centre of mass of each signal was determined using an in-house macro that utilised the Foci_Picker3D Plugin (Version 1.0, CAS, China[54]. Data were exported to Excel (Microsoft, USA) and distances between foci (separation) were calculated using 3D Pythagoras.

**Foci intensity measurement**. For analysis of MTOC intensities, z-stacks of the entire spindle region of oocytes expressing Cep192-GFP and H2B-mCherry were acquired (~40 × 40 × 37.5 μm, z-resolution 1.5 μm) with a 63× objective (NA 1.4). Using Image-J, sub-volumes containing the spindle poles were selected and an in-house macro utilising the ImageJ plugin Foci_Picker3D[54] was used to find the integrated intensity of all MTOCs and background subtracted at each pole. The same threshold was used for both poles of each spindle as the background of each pole was always observed to be the same.

To find the intensity of centromere and kinetochore proteins the 3D positions of all foci were logged using an in-house macro in a pair-wise manner such that both foci belonging to the same bivalent were assessed consecutively and the side facing one spindle pole was always logged first. In-house macros then analysed the volume (1.6, 1.6, 2.7 μm) around each foci and utilised the Foci_Picker3D plugin to estimate the integrated intensity. The same threshold was applied to each foci within a bivalent (Spc24, Min. Sat.) or was applied to each foci within an oocyte (Maj. Sat.).

**Spindle intensity measurements**. Oocytes expressing α-Tubulin-GFP and H2B-mCh were matured to 7 h after NEBD and the spindle image captured using a 40× objective, an xy pixel size of 0.1 μm and a z-step of 1 μm. Only oocytes where the long axis of the spindle was in the x–y plane were included such that centromere–kinetochore separation could be measured accurately. Spindles were analysed using Image J software, by creating a region of interest and using the 'plot profile' plugin to capture the tubulin and H2B intensity along the long axis of the spindle. A background profile was obtained likewise in a directly adjacent area. Data were exported to Microsoft Excel, background values were subtracted, and the centre of mass of the bivalents calculated. This was used to define the mid-point of the spindle. The sum of the tubulin intensities within 10 μm either side of the mid-point was calculated and normalised with respect to the cortical half of the spindle.

**Immunofluorescence microscopy**. Oocytes were fixed for 30 min in PBS containing 2% formaldehyde and 0.05% Triton-X, and were then permeabilised for 15 min in PBS containing 1% PVP and 0.05% Triton-X. Fixing and permeabilising was performed at room temperature and oocytes were extensively washed with PBS between stages. Oocytes were incubated at 4 °C overnight in a blocking buffer of 7% goat serum in PBS supplemented with 0.05% Tween-20. Primary antibody used was mouse anti-pericentrin (BD Biosciences 611815; 1:500)[32]. Secondary antibody used was goat anti-mouse Alexafluor 488 (Life Technologies, UK, 1:500, #a-21070). These incubations were at 37 °C in blocking solution. Chromatin was briefly counterstained with Hoechst (20 μg mL$^{-1}$) before imaging. For analysis of MTOC volume at two spindle poles, pericentrin was imaged as z-stacks with 1.5 μm spacing in fixed oocytes as for Cep192-GFP live analysis.

**Chromosome spreading**. We performed chromosome spreading as described previously[55]. Briefly, GV oocytes were released from milrinone arrest and matured to 5 h after NEBD. The zona pellucida was removed by brief treatment in acid Tyrode's solution and oocytes transferred to 5 μL of chromosome spreading

solution (1% paraformaldehyde, 0.15% triton X100, 3 mM DTT, in water) on a glass slide. Slides were then left to air dry at room temperature.

**FISH staining**. The spread chromosomes were cultured with 3 μL mouse chromosome 17 FISH probe (Zeiss Ltd, UK) on a heat block at 75 °C for 2 min then incubated in a humidified chamber at 37 °C overnight. Slides were washed in 0.4 × SSC (1 × SSC: 0.15 M NaCl, 0.015 M sodium citrate, pH 7.0) at 72 °C for 2 min and 2 × SSC (with 0.05% Tween 20) at room temperature for 30 s. Chromosomes were then stained with DAPI (20 mg mL$^{-1}$) for 10 min before imaging on a confocal microscope.

**Statistical analysis**. Sample means were compared with either a two-way Student's t-test, paired t-test, or one-way ANOVA with a post-hoc test as stated, following a test for normality using a Shapiro–Wilk test. Dichotomous data were compared using Fisher's exact test. All tests were performed using GraphPad Prism v7.0c (GraphPad Software, Inc).

**Data availability**. The data that support the findings of this study are available within the article and its Supplementary Information or from the corresponding authors upon reasonable request.

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

## Acknowledgements

The work was supported by a Leverhulme Trust grant (RPG-2017-352) to K.T.J., T.W. is supported by a China Scholarship Council studentship. The measurements of centromere–kinetochore separation were made through funding to K.T.J. from the BBSRC (BB/L006006/1). We thank Dr. Karen Schindler, Rutgers University, for the gift of the aurora kinase construct; Dr. Melina Schuh, Max Planck Institute for Biophysical Chemistry for the Cep192 construct, and Dr. Marie-Hélène Verlhac, College de France, CNRS for the tubulin construct.

## Author contributions

K.T.J. and S.I.L. devised the study. T.W., S.L.M. and S.I.L. performed the experiments and analysis. The manuscript was written by K.T.J. and S.I.L. with input from all authors.

## Additional information

**Competing interests:** The authors declare no competing interests.

