## [Peer Review File · Nature Communications]

Reviewers' comments:

Reviewer #1 (Remarks to the Author):

Oocytes divide by out-pocketing twice half of their DNA content into two tiny polar bodies. An important question is whether these asymmetric divisions in terms of size of daughter cells are also asymmetric in their DNA content. Such a phenomenon has been referred as "meiotic drive". Using oocytes from F1 hybrids mouse strain, presenting a difference in centromere size, authors present a transmission ratio distortion of 3:1 for two bivalents with smaller kinetochores. They propose a model in which meiotic drive is explained by the impact of microtubule force asymmetry, due to inherently biased spindle asymmetry, on chromosomes with different sized kinetochores. The work needs major revision prior publication.

1/ The absolute value for the measure of volumes for both major and minor satellite markers used to follow centromeres should appear in the manuscript, not just the ratio of volumes. In the methods, it is mentioned that these structures are followed using a 40X objective (the numerical aperture is not given). The Z resolution for the best 40X objective (NA =1.4) is 500nm. Based on images presented in Figure 1c and 1d, the volumes of the smallest major satellite region as well as of the minor satellite regions are close to the resolution limit of the objective. Hence, the ratio of volumes between small and large repeats is underestimated. So it is impossible to determine whether the measures of ratio make any sense (Figures 1 and 2). This constitutes a major issue.

2/ Authors should perform fish analysis to confirm the identification of chromosome 17 and 4 as the two chromosomes with specific organization of major and minor satellite regions. Without this and based on the main criticism addressed in point 1, conclusions from Fig 1c, d, e and f are extremely difficult to make.

3/ Treatment of oocytes with ZM447439, an Aurora B kinase inhibitor suggests that it participates in the asymmetry of segregation of chromosome 17 towards a more frequent extrusion of the larger major satellite-containing chromosome into the first polar body. However, treatment with this pan-Aurora kinase inhibitor results in defects in meiosis progression and chromosome alignment (Shuda Mol Reprod Dev 2000), so it is difficult to interpret results presented in Figure 2b and 2f. Other more specific tools to address the contribution of Aurora B in this process should be used: conditional knockout, antibody injection, dominant negative construct, etc...

4/ A recent study (Akera T, Science 2017) has shown the presence of an asymmetry in the spindle induced by the proximity to the cortex. Measures of MTOCs volume presented here suggest that the first meiotic spindle is already asymmetric 1-2h after NEBD, independently of spindle migration. Could authors discuss the discrepancy between their model and the model of Akera et al? How can they reconcile their data and the one of Akera, which did not observe spindle asymmetry early on?

5/ Authors do not give any potential explanation for the origin of MTOCs volume asymmetry, the basis of their new model.

Reviewer #2 (Remarks to the Author):

This manuscript investigates meiotic drive, biased chromosome segregation of selfish chromosomes into eggs during meiosis of female oocytes. First, the authors establish their experimental model using F1 hybrid mice (C57BL/6 x SJL), where they can distinguish maternal and paternal centromeres carrying different sizes of major satellite repeats in living oocytes. Using this model, they showed that the centromere with more major satellites is preferentially

segregated into the polar body. Their beautiful chromosome tracking showed that when the centromere with more major satellites initially faced the oocyte center, the chromosome preferentially underwent rotation to reorient the centromere with more major satellites towards the cortex. Chromosome rotation is expected to involve dynamic exchanges of kinetochore-microtubule attachments. Consistently, the authors show that the rotation depended on Aurora kinases, attachment destabilizing factors. Their careful quantitative analysis also detected asymmetric distribution of spindle components. They show that tubulin density and MTOC volume on the meiotic spindle are greater on the spindle half directing towards the cortex. This spindle asymmetry may provide possible explanation for preferential positioning of the stronger centromere-kinetochore on the spindle half directing towards the cortex.

Overall, this manuscript provides excellent observation of the behaviour of chromosomes and the spindle that lead to biased chromosome segregation. Their chromosome tracking analysis is beautiful. They also nicely show spindle asymmetry through careful quantification of MTOC and tubulin signals. My major concern is that the manuscript is largely descriptive and lacks functional assays, except the Aurora inhibition experiment that yielded expected results. The manuscript would be significantly improved by adding experiments to address the functional significance of tubulin or MTOC asymmetry in biased chromosome segregation. Such experiments are essential to test their model and would be feasible if they could reveal molecular mechanisms underlying the tubulin and MTOC asymmetry.

Major comments

1. The manuscript is entitled 'Spindle and kinetochore-centromere asymmetry cause meiotic drive in oocytes'. I am afraid that this title does not highlight new points revealed by this manuscript. Even before this study, it is theoretically clear that meiotic drive can be accomplished only when both the spindle and kinetochores have some kind of asymmetry. Moreover, the manuscript did not address the causal relationship between the asymmetric tubulin density or MTOC volume with meiotic drive. I would suggest that the authors consider a title that emphasises their finding of tubulin density and MTOC asymmetry that give a possible explanation to meiotic drive.
2. It would be very surprising to me if the segregation of chromosome 17 and 4 would be biased as extremely as they show in their experimental system (nearly 8:2 ratio!). One concern would be an artefact of Mj.Sat.-mClover. This surprising ratio should be confirmed by immunostaining or other methods.
3. I was confused by the fact that the stronger centromere (more major satellites) is preferentially segregated to the polar body. Is this opposite to what was described in Chmatal et al 2014 and Akera et al 2017? If so, it would be helpful to discuss possible explanations for this apparent discrepancy in the manuscript. Again, this might be due to an artefact of Maj.Sat.-mClover.

Minor points

1. I think they used ZM447439 as a pan-Aurora inhibitor, as written in the main text, but in some figures it is written as Aurora B/C inhibitor (Figure 2f). Was ZM447439 used to inhibit Aurora A/B/C or B/C? This depends on the concentration they used, but I could not find this information in the manuscript.
2. Figure 3e requires more detailed information of the experimental procedure. Was Cytochalasin added after or before spindle migration?

Reviewer #3 (Remarks to the Author):

This manuscript makes a valuable contribution by introducing a new system to investigate female meiotic drive in mammals. The results show that in BL6/SJL hybrid mice the centromere with less major satellite preferentially remains in the egg by re-orienting on the spindle to preferentially attach to the side facing away from the cortex. The findings represent an important advance because the field is severely limited by the lack of experimental systems to study the cell biology of meiotic drive. The more mechanistic claims made in this paper, however, are overstated. The title states that "spindle and kinetochore-centromere asymmetry cause meiotic drive in oocytes". The results show correlations but not any causal relationships. The abstract states that "a greater concentration of microtubule organizing centres at the cortical pole ... affects tubulin density and causes a measurable increase in the pulling forces". The observations are correlative, and there is no experiment to test causality. The last paragraph states that "our data present a mechanistic basis for meiotic drive", but the paper provides little mechanistic insight. The data show interesting differences in satellite DNA and kinetochore protein levels and in centromere-kinetochore distance between the two sides of the bivalent, but it is not clear how any of these differences bias the segregation. The asymmetry in the spindle is also intriguing, but there is no experiment to test whether this asymmetry has any functional significance for meiotic drive. Overall, I recommend publication because of the importance of characterizing a new meiotic drive system, with significant revisions to the text.

Additional comments:

1. The authors suggest that minor satellite DNA may influence segregation by transcription of "non-coding RNAs that influence Aurora kinase activity" (p. 3-4). This model seems to require that the RNAs act locally at the centromeres from which they are produced. Is there evidence for such a phenomenon? This point should be discussed.
2. Fig. 2 shows centromere orientation relative to the cortex. The spindle is in the center of oocyte (i.e., before migration to the cortex) for the "initial" orientation measurement, and it is unclear how the authors defined the cortical direction at this point. Similarly, there are other parts of the manuscript where it is not clear how the cortical and central sides of the spindle were defined before spindle migration (p. 6 for example). This point is important because interpretation of the bivalent rotation and spindle asymmetry findings depends on it.
3. The authors' use of the term "tension" is confusing. Dating back to the classic experiments of Nicklas (also in meiosis I), tension typically refers to spindle forces pulling the two sides of the bivalent in opposite directions. In this paper it is used to refer to distance between the major satellite repeats and the outer kinetochore. The authors should pick a different word to avoid confusion.
4. The text states that "the spindle appears to exert an asymmetrical force on the bivalents" (p. 7). The wording of this statement is confusing. Force is a vector with a magnitude and direction. What does "asymmetrical force" mean? If it means that one side of the spindle exerts more force than the other side, then the bivalents should move in the direction dictated by the greater force. If the bivalents are moving, then the data should show it. If they are not moving, then the forces are not asymmetric.
5. In several places volume is measured from fluorescence images (e.g., major satellite repeat volume on p. 3 and MTOC volume on p. 8). It's not clear why volume is the relevant measurement, rather than integrated intensity. The authors should either measure the integrated intensity, which would indicate relative amounts of major sat repeats or spindle pole proteins, or explain why volume is a better choice.
6. The authors may want to consider citing previous observations of spindle pole asymmetry:

Carabatsos et al. 2000 (*Microsc Res Tech* 49: 435-44) and Michaut et al. 2005 (*Dev Biol* 280: 26-37).

7. If spindle asymmetry arises from difference between the poles, as the authors suggest, this asymmetry should be present at early stages of meiosis I (by 4 h according to Fig 4g). Otherwise, other factors may contribute.

8. The data presentation in Fig. 1 c-e is confusing. The legend states that major sat is in green, minor sat in red, and Spc24 in magenta. Panel d (labeled minor sat) shows blue and green but no red, which does not match the legend. Rather than using color, it would be better to show greyscale images of each channel side-by-side so that one does not obscure the other. Also, how was minor satellite labeled in Fig. 1d? The methods (p. 16) refer to a single plasmid from Addgene for both major and minor satellite. Presumably there are different plasmids for the two different classes of satellite repeats.

9. A revised manuscript should discuss the findings in the context of the recently published study by Akera et al. (which was likely published after this manuscript was submitted) that addresses spindle asymmetry and meiotic drive.

We thank the reviewers for their helpful suggestions and comments. You can see that the manuscript has undergone extensive revision with the inclusion of further experiments. Here we give a detailed response to the points raised. Please note additionally we have made a few minor formatting changes to comply with Nature Communications guidelines.

Reviewer 1

1/ The absolute value for the measure of volumes for both major and minor satellite markers used to follow centromeres should appear in the manuscript, not just the ratio of volumes. In the methods, it is mentioned that these structures are followed using a 40X objective (the numerical aperture is not given). The Z resolution for the best 40X objective (NA =1.4) is 500nm. Based on images presented in Figure 1c and 1d, the volumes of the smallest major satellite region as well as of the minor satellite regions are close to the resolution limit of the objective. Hence, the ratio of volumes between small and large repeats is underestimated. So it is impossible to determine whether the measures of ratio make any sense (Figures 1 and 2). This constitutes a major issue.

We realise that the resolution of our, or any, confocal system will not be able to accurately measure volumes where the specimen size is on a similar scale to the axial resolution. In addition to this there is inherent variability in the injection volume of the TALE mRNA, leading to variability in signal intensity at the centromere. Thus measures of absolute volumes are not appropriate, especially when comparing between oocytes. However, a ratio between two centromeres of the same bivalent, using the same imaging parameters allows us to draw a conclusion about the relative signal. We have now switched to using the integrated intensity, since the volume is subject to the mRNA injection volume and the threshold used to define cut-off for volume measurement.

2/ Authors should perform fish analysis to confirm the identification of chromosome 17 and 4 as the two chromosomes with specific organization of major and minor satellite regions. Without this and based on the main criticism addressed in point 1, conclusions from Fig 1c, d, e and f are extremely difficult to make.

We have now addressed this issue by using FISH staining for chromosome 17 (Fig S2). This driving bivalent is the main focus of the work. Unfortunately, we did not get staining for chromosome 4 to work- but this chromosome is only studied in Fig1. Using this staining we are able to show that chromosome 17 does indeed have very strong asymmetry within the centromere region.

3/ Treatment of oocytes with ZM447439, an Aurora B kinase inhibitor suggests that it participates in the asymmetry of segregation of chromosome 17 towards a more frequent extrusion of the larger major satellite-containing chromosome into the first polar body. However, treatment with this pan-Aurora kinase inhibitor results in defects in meiosis progression and chromosome alignment (Shuda Mol Reprod Dev 2000), so it is difficult to interpret results presented in Figure 2b and 2f. Other more specific tools to address the contribution of Aurora B in this process should be used: conditional knockout, antibody injection, dominant negative construct, etc, etc...

We thank the reviewer for this suggestion. We have repeated the experiment following over-expression of a dominant negative aurora C isoform (T210A,T214A) which also blocks Aurora B (Balboula & Schindler PLoS Genet. 2014 10(2):e1004194). We find that this dominant negative AurB/C construct has the same effect as ZM447439 (see , Results “Meiotic drive is dependent on aurora kinase activity”; Fig 2f).

4/ A recent study (Akera T, Science 2017) has shown the presence of an asymmetry in the spindle induced by the proximity to the cortex. Measures of MTOCs volume presented here suggest that the first meiotic spindle is already asymmetric 1-2h after NEBD, independently of spindle migration. Could authors discuss the discrepancy between their model and the model of Akera et al? How can they reconcile their data and the one of Akera, which did not observe spindle asymmetry early on?

We have now looked at spindles at 4 hours after NEBD. We find that the spindle asymmetry can be detected here too, as predicted by the MTOC asymmetry (new Fig 6). The asymmetry is slight at this timepoint but is accompanied by robust measures of MTOC asymmetry and tension asymmetry across bivalents. Further, the 4h timepoint coincides with the driving bivalent re-orientation. We speculate that the asymmetry becomes more pronounced as the spindle matures and moves within proximity of the cortex, but note that biased selection of the asymmetric chromosome has already taken place prior to spindle migration. The work of Dr Lampson's lab (Akera et al) is now included in the discussion. As detailed in this section our study observes drive at an earlier timepoint (ie before spindle migration).

5/ Authors do not give any potential explanation for the origin of MTOCs volume asymmetry, the basis of their new model.

It may well be that the cortical to central movement of the MTOCs prior to NEBD (Fig 6b) puts in place an asymmetry that is maintained even though the MTOCs flatten and disperse at NEBD. In the Discussion we state a possible explanation: "The reason for the establishment of MTOC asymmetry remains unknown, but could relate to the much earlier journey MTOCs make from the oocyte cortex to the nuclear envelope before NEB. It may well prove that MTOCs clustering at this time of NEBD mark an orientation for future migration of the spindle."

Reviewer 2

1.. I would suggest that the authors consider a title that emphasises their finding of tubulin density and MTOC asymmetry that give a possible explanation to meiotic drive.

Thank you- we have changed the title to "Spindle tubulin and MTOC asymmetries could explain meiotic drive in oocytes".

2. It would be very surprising to me if the segregation of chromosome 17 and 4 would be biased as extremely as they show in their experimental system (nearly 8:2 ratio!). One concern would be an artefact of Mj.Sat.-mClover. This surprising ratio should be confirmed by immunostaining or other methods

The reviewer thinks that our reported drive rates (~ 75%) is an extreme measure. However the reviewer should be aware of similar and higher rates of drive in other systems. An extreme TRD would be 98% as seen in monkeyflower hybrids (Fishman and Saunders, Science 322:1559-1562, 2008). B -chromosomes in grasshopper oocytes are at a similar level to ours (78%; Hewitt, Chromosoma 56: 381-391 1976)); and in mouse oocytes carrying a univalent X chromosome there is also a comparable rates of drive (67%; LeMaire-Adkins and Hunt, Genetics 156:775-783, 2000). Therefore there is no evidence based on the rate of drive that the Maj-Sat Clover has any influence on this process.

3. I was confused by the fact that the stronger centromere (more major satellites) is preferentially segregated to the polar body. Is this opposite to what was described in Chmatal et al 2014 and Akera et al 2017? If so, it would be helpful to discuss possible explanations for this apparent discrepancy in the manuscript. Again, this might be due to an artefact of Maj.Sat.-mClover.

We apologise for the confusion which is now clarified in the Discussion. The present work supports the 'strong centromere' hypothesis. See Discussion first two paragraphs. The papers by the Lampson group used a cenp-B marker, which corresponds to the minor satellite region, closer to the kinetochore than the major satellite. We show in Fig 1 that a smaller Min.Sat signal is associated with a larger Maj.Sat signal.

4. I think they used ZM447439 as a pan-Aurora inhibitor, as written in the main text, but in some figures it is written as Aurora B/C inhibitor (Figure 2f). Was ZM447439 used to inhibit Aurora A/B/C or B/C? This depends on the concentration they used, but I could not find this information in the manuscript.

We have added the dose we used, 10 μ M ZM (a dose we have used previously; Lane et al Reproduction 2010) to the methods section. The IC50 for aurora A and B is very similar (Ditchfield et al., 2003 161(2):267-80). Therefore we should have been clear to identify this as a pan-aurora inhibitor. We have done so in the revised text. Additionally, by request from another reviewer, we have repeated these experiments with an Aurora B/C DN mutant (See p6).

5. Figure 3e requires more detailed information of the experimental procedure. Was Cytochalasin added after or before spindle migration?

Cytochalasin B was added at 6 h after NEBD (so spindle migration was advanced enough to distinguish centre/cortex). Measurement was made at 7 h after NEBD. We have added more detail to the methods (This figure is now Fig 4d).

Reviewer 3

1. The more mechanistic claims made in this paper, however, are overstated. The title states that “spindle and kinetochore-centromere asymmetry cause meiotic drive in oocytes”. The results show correlations but not any causal relationships. The abstract states that “a greater concentration of microtubule organizing centres at the cortical pole ... affects tubulin density and causes a measurable increase in the pulling forces”. The observations are correlative, and there is no experiment to test causality. The last paragraph states that “our data present a mechanistic basis for meiotic drive”, but the paper provides little mechanistic insight. The data show interesting differences in satellite DNA and kinetochore protein levels and in centromere-kinetochore distance between the two sides of the bivalent, but it is not clear how any of these differences bias the segregation. The asymmetry in the spindle is also intriguing, but there is no experiment to test whether this asymmetry has any functional significance for meiotic drive.

We have now modified the manuscript to ensure we do not overstate causality. The title and abstract have been changed to reflect this. The observations presented in the work help present a model to be tested in future work. The paper shows a relationship between Aurora kinase activity and meiotic drive. They also show in detail timing of the events of drive not previously obtained.

2. The authors suggest that minor satellite DNA may influence segregation by transcription of “non-coding RNAs that influence Aurora kinase activity” (p. 3-4). This model seems to require that the RNAs act locally at the centromeres from which they are produced. Is there evidence for such a phenomenon? This point should be discussed.

We include two references to work that demonstrate the ability of non-coding centromeric mRNAs to influence aurora activity or chromosome stability. Chan et al., 2017 and Ferri et al., 2009. There is an assumption that similar to Xist acting on the X-chromosome the interaction is in-cis (See Discussion in Ferri et al, 2009). However we note this remains to be proved. We highlight this fact in the revised text (fourth paragraph Discussion, beginning “Additionally).

3. Fig. 2 shows centromere orientation relative to the cortex. The spindle is in the centre of the oocyte (i.e., before migration to the cortex) for the “initial” orientation measurement, and it is unclear how the authors defined the cortical direction at this point. Similarly, there are other parts of the manuscript where it is not clear how the cortical and central sides of the spindle were defined before spindle migration (p. 6 for example). This point is important because interpretation of the bivalent rotation and spindle asymmetry findings depends on it.

For Fig 2 we explain that we were able to define cortical versus central because we had captured an entire timeseries and were able to track the bivalent in each frame. Hence our initial position was an extrapolation back from the future direction of travel. See revised text p5. However in Fig 4c, at the 4h

timepoint, we do not have future extrapolation so in the revised manuscript we do not use 'cortical' or 'central'. Here we refer to the two sides of the C-Kt as having greater or lesser separation.

4. The authors' use of the term "tension" is confusing. Dating back to the classic experiments of Nicklas (also in meiosis I), tension typically refers to spindle forces pulling the two sides of the bivalent in opposite directions. In this paper it is used to refer to distance between the major satellite repeats and the outer kinetochore. The authors should pick a different word to avoid confusion.

We now use the words separation or stretch to refer to the distance between centromere and kinetochore throughout the text and avoid use of the word tension. We are of course measuring distances and using this as a proxy for tension, working under the assumption that the material being measured is elastic. We now explain this assumption and provide reference in the text (p8 last paragraph onwards) .

5. The text states that "the spindle appears to exert an asymmetrical force on the bivalents" (p. 7). The wording of this statement is confusing. Force is a vector with a magnitude and direction. What does "asymmetrical force" mean? If it means that one side of the spindle exerts more force than the other side, then the bivalents should move in the direction dictated by the greater force. If the bivalents are moving, then the data should show it. If they are not moving, then the forces are not asymmetric.

The reviewer is correct in that the bivalents do not appear to move towards one end of the spindle, instead staying in the centre of the spindle. We believe this is because in addition to increased pulling forces (inferred from intra-kinetochore stretch) there are likely to be increased pushing forces in equal measure, for example the polar ejection force. We have noted an increased tubulin density on the side of the spindle displaying increased C-Kt stretch and therefore it seems reasonable that there may be an increase in both K-fibre and non-k-fibre microtubules acting on one side of the bivalent. We therefore believe that although C-Kt stretch, and by inference kinetochore derived tension, are asymmetrical across the bivalent, there is no net force acting on the bivalent which would cause it to move polewards.

6. In several places volume is measured from fluorescence images (e.g., major satellite repeat volume on p. 3 and MTOC volume on p. 8). It's not clear why volume is the relevant measurement, rather than integrated intensity. The authors should either measure the integrated intensity, which would indicate relative amounts of major sat repeats or spindle pole proteins, or explain why volume is a better choice.

Done. We have now switched to using integrated intensity throughout the manuscript.

7. The authors may want to consider citing previous observations of spindle pole asymmetry: Carabatsos et al. 2000 (Microsc Res Tech 49: 435-44) and Michaut et al. 2005 (Dev Biol 280: 26-37).

Done. Thank you for alerting us to these 2 relevant papers, which are now cited (p10).

8 If spindle asymmetry arises from difference between the poles, as the authors suggest, this asymmetry should be present at early stages of meiosis I (by 4 h according to Fig 4g). Otherwise, other factors may contribute.

Done. We have performed this analysis at 4h after NEBD, and show that tubulin asymmetry does exist (Fig 6 i,j,k).

9. The data presentation in Fig. 1 c-e is confusing. The legend states that major sat is in green, minor sat in red, and Spc24 in magenta. Panel d (labelled minor sat) shows blue and green but no red, which does not match the legend. Rather than using color, it would be better to show greyscale images of each channel side-by-side so that one does not obscure the other. Also, how was minor satellite labeled in Fig. 1d? The methods (p. 16) refer to a single plasmid from Addgene for both major and minor satellite. Presumably there are different plasmids for the two different classes of satellite repeats.

Done. We have revised Fig 1 (now 1c) to address these comments. We have also revised the Methods section (see 'crRNA manufacture') to state both Addgene catalogue numbers.

10. A revised manuscript should discuss the findings in the context of the recently published study by Akera et al. (which was likely published after this manuscript was submitted) that addresses spindle asymmetry and meiotic drive.

Done. Akera et al is now cited widely in the revised text, with a focus on it relative to the present work in the Discussion, paragraph beginning "For meiotic drive to work in favour of retaining homolog pairs..."

Reviewers' comments:

Reviewer #1 (Remarks to the Author):

Authors have revised their work, and have address some of the reviewers comment. In particular, they now present measure of intensities rather than volume for major and minor satellite asymmetries, which is much better. However there are still major caveats in the revised work that preclude publication in Nature Communications.

Major point

1/ authors never mention in any figure legends the number of independent experiments and rarely the number of oocytes analysed. This should be mentioned since some observations appear to come from very small number of samples (Fig 1c, Fig 3b, Fig 5b, Fig 6 a).

2/ Authors should tone down their evidence for meiotic drive since as presented in Fig 2b, the major satellite bias towards the cortex is barely significant (p value 0.016).

3/ Authors should show movies associated with the measure of meiotic drive (Fig 2c) so that the reader can grasp more easily how these asymmetries were followed.

4/ Data presented in Figure 3 have been already extensively published. This figure should be removed and authors should simply mention in the text that asymmetric stretch on bivalents is established before spindle migration. Furthermore, authors should cite previous work. Indeed, Verlhac CB 2000 already showed 18 years ago that the first meiotic spindle starts its migration around 6.5h after NEBD (see authors lane 191). Also, the speed of spindle motion was precisely measured before (mean speed: 84 nm.min⁻¹ from Verlhac CB 2000; 100 nm.min⁻¹ from Schuh & Ellenberg CB 2008 and 130 nm.min⁻¹ from Li NCB 2008). It is slightly dishonest to pretend nothing was done. Moreover, previous works are consistent with observations made here, where the mean speed is measured between 70 to 100 nm.min⁻¹ (Lane 194 of the manuscript).

5/ Measure of the asymmetric stretch on bivalents (Fig 4) is problematic. Here again, authors measure, what they call 3D distances (lane 481 of the manuscript) below the Z resolution of their microscope as shown on Fig c and d (about 500 nm). This figure should be removed.

6/ Measure of the intensity of tubulin-GFP on the spindle is also problematic (Fig 5). First: as presented in their example (Fig 5a), the chromosomes are not symmetrically distributed along what is defined as the middle spindle axis (pink lane). Thus one can observe holes on the left side of the tubulin-GFP staining, corresponding to the location of chromosomes, which are not observed on the right side. The staining is thus not homogenous, therefore barely amenable to quantification. Second: as shown on Fig 5b, the basal level (background) fluorescence is higher on the right side of the spindle (blue region) compared to left side (orange region). This is due to optics of the specimen almost neglected here: light from the objective will illuminate differently the centre of a 80 micron-wide cell than the cortex (absorption, diffraction). Therefore, quantifying these two sides makes little sense. Third: no statistics are being presented on these measures (Fig 5c). Authors should measure whether the variance of intensity of fluorescence inside each compartment (left or right) is actually lower than the variance between the two compartments. For this, more than 10 oocytes should be observed. Fourth: spindles are rarely parallel to the plane of observation and any tilt will compromise the measure of left and right compartments. To circumvent this issue, authors should sum-up multiple Z as well as increase the number of spindles observed. However from the methods, it is not clear how intensities were measured. Was it on a maximal projection? If so from how many Z planes?

7/ Same problems as the ones raised in point 6 apply to measures of MTOC intensities.

8/ Authors should remove Fig 6a since they are showing MTOC motion from the cortex to nucleus

in an incompetent oocyte (presenting non-surrounded nucleolus) instead of a competent one (surrounded nucleolus). As published before, but not even cited here (Luksza Dev Biol 2013), MTOCs migrate from the cortex to the nuclear periphery during oocyte growth, then they undergo fragmentation after meiosis resumption, as first observed in (Luksza Dev Biol 2013).

Reviewer #2 (Remarks to the Author):

My concerns have been fully addressed in the revised manuscript. The manuscript is still descriptive and lacks mechanical insight, but the authors appropriately discuss their model as a hypothesis. Overstatements that were found in the previous version of the manuscript have been removed.

Reviewer #2's additional comments on Reviewer #1's major points (corresponds to Reviewer #1's "Major points")

1. I agree with the reviewer 1 that the authors should mention the number of independent experiments. To me the n-values of oocytes are appropriately described in the manuscript or shown in the figure itself, and the statistical analysis seems to be appropriately done, which well supports their conclusion.

2. I understand the reviewer 1's point that the p-value is low in this particular experiment. The authors would be encouraged to increase the n-value of oocytes by additional experiments for this particular figure. However I am already convinced with the result, because the data in Fig 2b is supported by Fig 1c and Fig 2e,f.

3. I agree.

4. In my opinion Figure 3 should be presented in this paper. The timing of spindle migration can be affected by mouse strains and culture conditions. It is important that the timing of spindle migration is determined in the experimental condition that the authors used for this paper. I agree with the reviewer 1 that the authors should cite previous works and could remove unnecessary descriptions such as spindle migration speeds.

5.

For clarification, the description about 3D measurement in Fig 4 is found in the lane 462-472. The lane 481 explains the measurement of MTOC volumes, which is not in Fig 4.

Based on the description in the lane 462-472, the z-interval used for imaging in Fig 4 was 300nm, which would not be sufficient to robustly resolve centromere-kinetochore separation (~500nm) if the separation was along z-axis. But I would assume that the authors did this analysis using images in which sister-kinetochore-axis well parallel to the focal plane, which has pixel size of 35nm in xy – that would be sufficient to resolve centromere-kinetochore separation. Based on this assumption I would be convinced with the result. The authors should add further explanations in the Method.

6. For the first point, authors should provide a better representative image. The image shown in Fig 5a appears to have H2B off-centred, but the authors determined the mid-point of the spindle based on the centre-of-mass of chromosome signals (lane 489-500), so quantification results should be valid.

For the second point, the authors appropriately subtracted background intensity (lane 489-500), so the contribution of background difference would be minimum.

For the third point, it would be nice if the authors could statistically show the significance in the tubulin signal asymmetry, as the reviewer 1 suggests. I am almost convinced that it is significant as all the values presented in Fig 5c are above 1.0.

For the fourth point, the authors should add descriptions to the Method to clarify whether they used z-slices for the measurement. I would not think sum-up of intensities from multiple z-slices is essential in this case – it would be more important that the measurement ROIs for left and right should be positioned symmetrically with respect to the metaphase plate in 3D.

7. In the manuscript (lane 473-), it is clearly described that the authors measured integrated intensities in 3D for MTOCs. The authors should add descriptions about background subtraction.

8. As the reviewer 1 points out, the oocyte shown in Fig 6a appear to be a non-surrounded-nucleolus oocyte, which is unlikely to be meiotically competent. Do the authors have a better representative image? It would be also important to check if Fig 6b included data from non-surrounded-nucleolus oocytes.

Response to Reviewers

Reviewer #1

1/ authors never mention in any figure legends the number of independent experiments and rarely the number of oocytes analysed. This should be mentioned since some observations appear to come from very small number of samples (Fig 1c, Fig 3b, Fig 5b, Fig 6 a).

Done. See Figure legends, independent repeat numbers for all experiments are now included in the legends.

2/ Authors should tone down their evidence for meiotic drive since as presented in Fig 2b, the major satellite bias towards the cortex is barely significant (p value 0.016).

It is not correct to state that $P=0.016$ is 'barely significant'. By convention, statistical significance is claimed when $P<0.05$. No changes are deemed necessary.

3/ Authors should show movies associated with the measure of meiotic drive (Fig 2c) so that the reader can grasp more easily how these asymmetries were followed.

Done. Movie now included (See Movie 1 in SI).

4/ Data presented in Figure 3 have been already extensively published. This figure should be removed and authors should simply mention in the text that asymmetric stretch on bivalents is established before spindle migration. Furthermore, authors should cite previous work. Indeed, Verlhac CB 2000 already showed 18 years ago that the first meiotic spindle starts its migration around 6.5h after NEBD (see authors lane 191). Also, the speed of spindle motion was precisely measured before (mean speed: 84 nm.min⁻¹ from Verlhac CB 2000; 100 nm.min⁻¹ from Schuh & Ellenberg CB 2008 and 130 nm.min⁻¹ from Li NCB 2008). It is slightly dishonest to pretend nothing was done. Moreover, previous works are consistent with observations made here, where the mean speed is measured between 70 to 100 nm.min⁻¹ (Lane 194 of the manuscript).

We have revised the text to explain the reasoning for Fig 3, and added your citations. "Although previous studies have defined the timing of spindle migration as well as details of its speed, these parameters may vary from strain to strain. As such it was important to make such measurements in the present study 19–21." We had not meant to imply that this was the first time such measurements had been made, and apologise for the inference drawn from not stating this directly.

5/ Measure of the asymmetric stretch on bivalents (Fig 4) is problematic. Here again, authors measure, what they call 3D distances (lane 481 of the manuscript) below the Z resolution of their microscope as shown on Fig c and d (about 500 nm). This figure should be removed.

We have explained our measurements as suggested by Reviewer 2 in the revised Methods (whose points on how we measure are indeed correct). We do not feel therefore there is any need to remove this figure.

6/ Measure of the intensity of tubulin-GFP on the spindle is also problematic (Fig 5). First: as presented in their example (Fig 5a), the chromosomes are not symmetrically distributed along what is defined as the middle spindle axis (pink lane). Thus one can

observe holes on the left side of the tubulin-GFP staining, corresponding to the location of chromosomes, which are not observed on the right side. The staining is thus not homogenous, therefore barely amenable to quantification. Second: as shown on Fig 5b, the basal level (background) fluorescence is higher on the right side of the spindle (blue region) compared to left side (orange region). This is due to optics of the specimen almost neglected here: light from the objective will illuminate differently the centre of a 80 micron-wide cell than the cortex (absorption, diffraction). Therefore, quantifying these two sides makes little sense. Third: no statistics are being presented on these measures (Fig 5c). Authors should measure whether the variance of intensity of fluorescence inside each compartment (left or right) is actually lower than the variance between the two compartments. For this, more than 10 oocytes should be observed. Fourth: spindles are rarely parallel to the plane of observation and any tilt will compromise the measure of left and right compartments. To circumvent this issue, authors should sum-up multiple Z as well as increase the number of spindles observed. However from the methods, it is not clear how intensities were measured. Was it on a maximal projection? If so from how many Z planes?

Please see the responses we made to these points for Reviewer 2 (point 6) below.

7/ Same problems as the ones raised in point 6 apply to measures of MTOC intensities.

Please see the responses we made to this for Reviewer 2 (point 7) below.

8/ Authors should remove Fig 6a since they are showing MTOC motion from the cortex to nucleus in an incompetent oocyte (presenting non-surrounded nucleolus) instead of a competent one (surrounded nucleolus). As published before, but not even cited here (Luksza Dev Biol 2013), MTOCs migrate from the cortex to the nuclear periphery during oocyte growth, then they undergo fragmentation after meiosis resumption, as first observed in (Luksza Dev Biol 2013).

Please see the responses we made to this for Reviewer 2 (point 8) below.

Reviewer #2

1/ I agree with the reviewer 1 that the authors should mention the number of independent experiments. To me the n-values of oocytes are appropriately described in the manuscript or shown in the figure itself, and the statistical analysis seems to be appropriately done, which well supports their conclusion.

Done. See revised Figure legends: independent repeat numbers for all experiments are now included in the legends.

2/ I understand the reviewer 1's point that the p-value is low in this particular experiment.

The authors would be encouraged to increase the n-value of oocytes by additional experiments for this particular figure. However I am already convinced with the result, because the data in Fig 2b is supported by Fig 1c and Fig 2e,f.

Thank you for this clarity. We respectfully remind the reviewer we have reached a level of significance for this of $p=0.016$

3/ I agree.

Done. Movie now included (see Movie 1 in SI)

4/ In my opinion Figure 3 should be presented in this paper. The timing of spindle migration can be affected by mouse strains and culture conditions. It is important that the timing of spindle migration is determined in the experimental condition that the authors used for this paper. I agree with the reviewer 1 that the authors should cite previous works and could remove unnecessary descriptions such as spindle migration speeds.

We have revised the text and added the citations of Reviewer 1. "Although previous studies have defined the timing of spindle migration as well as details of its speed, these parameters may vary from strain to strain. As such it was important to make such measurements in the present study^{19–21}."

5/ For clarification, the description about 3D measurement in Fig 4 is found in the lane 462-472. The lane 481 explains the measurement of MTOC volumes, which is not in Fig 4. Based on the description in the lane 462-472, the z-interval used for imaging in Fig 4 was 300nm, which would not be sufficient to robustly resolve centromere-kinetochore separation (~500nm) if the separation was along z-axis. But I would assume that the authors did this analysis using images in which sister-kinetochore-axis well parallel to the focal plane, which has pixel size of 35nm in xy – that would be sufficient to resolve centromere-kinetochore separation. Based on this assumption I would be convinced with the result. The authors should add further explanations in the Method.

Done. The reviewer's assumption is correct and we have added this detail to the Centromere-kinetochore separation section of Material and Methods. We should have made it clear that it would not be possible to reach this resolution along the z-axis, but should have included this so as to avoid any reader ambiguity.

6. For the first point, authors should provide a better representative image. The image shown in Fig 5a appears to have H2B off-centred, but the authors determined the mid-point of the spindle based on the centre-of-mass of chromosome signals (lane 489-500), so quantification results should be valid.

Fig 5a has been redrawn to show the correct Centre of Gravity. Mistakenly, and not picked up by us for which we apologise, it had moved by one or two pixels during the figure composition, which could be seen by eye. It has now been recentred. Note this has in no way affected the mid-point calculation in the more substantive Figure part (Fig 5b).

For the second point, the authors appropriately subtracted background intensity (lane 489-500), so the contribution of background difference would be minimum.

Agreed, thank you.

For the third point, it would be nice if the authors could statistically show the significance in the tubulin signal asymmetry, as the reviewer 1 suggests. I am almost convinced that it is significant as all the values presented in Fig 5c are above 1.0.

In the revised figure legend (Fig 5, as well as Fig 6) we have made it clear that the error bars represent 95% confidence intervals. Because the 95% CI range does not pass through 1.0 by convention it is $P < 0.05$ (i.e. it has reached a level of significance, $P < 0.05$). We should have made it clearer in the original legend what the error bars show and therefore that the data show a significantly different ratio from 1.0.

For the fourth point, the authors should add descriptions to the Method to clarify whether they used z-slices for the measurement. I would not think sum-up of intensities from multiple z-slices is essential in this case – it would be more important that the measurement ROIs for left and right should be positioned symmetrically with respect to the metaphase plate in 3D.

We have modified the Methods (Spindle intensity measurements) to make the methodology more clear. It now reads “Oocytes expressing α -Tubulin-GFP and H2B-mCh were matured to 7 h after NEBD and the spindle image captured using a 40x objective, an xy pixel size of 0.1 μ m and a z-step of 1 μ m to encompass the entire spindle. Tubulin intensities were then summed in the z-axis (a z-projection). Only oocytes where the long axis of the spindle was in the x-y plane were included.”

7. In the manuscript (lane 473-), it is clearly described that the authors measured integrated intensities in 3D for MTOCs. The authors should add descriptions about background subtraction.

We have modified the Methods (Foci Intensity Measurements) to make the methodology more clear. It now reads “Using Image-J, sub-volumes containing the spindle poles were selected and an in-house macro utilizing the ImageJ plugin Foci_Picker3D was used to find the integrated intensity of all MTOCs and background subtracted at each pole. The same threshold was used for both poles of each spindle as the background of each pole was always observed to be the same.”

8. As the reviewer 1 points out, the oocyte shown in Fig 6a appear to be a non-surrounded-nucleolus oocyte, which is unlikely to be meiotically competent. Do the authors have a better representative image? It would be also important to check if Fig 6b included data from non-surrounded-nucleolus oocytes.

The z -projection of the oocyte gave the wrong impression: it was SN. All oocytes used were SN, where chromatin staining was performed. We have added an insert to Fig 6a to show the single z-slice containing the nucleolus and have revised the text “In this experiment and all others where measurable, we used GV stage oocytes that had a Surrounded Nucleolus (SN) configuration for chromatin (Fig 6a) as these show the greatest potential for embryonic development, and likely represent a more mature state than the alternative non-surrounded nucleolus configuration”. However, we would respectfully point out, as shown in this reference (Ref. 37) SN and NSN oocytes both have the capacity to complete meiosis normally, but the NSN configuration tends to block in embryonic development. So it may not be so correct to say NSN oocytes are incompetent for completing meiotic maturation (this is probably true of only very small sized oocytes). We agree though that maturation rates of NSN oocytes are lower.